# Antioxidant Profile and Sensory Analysis in Olive Oils of Different Quality Grades

**Maria Teresa Frangipane** [1,*] , **Lara Costantini** [2] , **Nicolò Merendino** [2] and **Riccardo Massantini** [1,3]

1 Department for Innovation in Biological, Agro-Food and Forest Systems (DIBAF), University of Tuscia, via San Camillo de Lellis, 01100 Viterbo, Italy
2 Department of Ecological and Biological Sciences (DEB), Tuscia University, Largo dell'Università snc, 01100 Viterbo, Italy
3 Study Alpine Centre, University of Tuscia, Via Rovigo, 7, 38050 Pieve Tersino, Italy
* Correspondence: mtfrangi@unitus.it

**Abstract:** Antioxidant capacity and sensory analysis of olive oils of different quality grades (Extra virgin, Virgin, Ordinary and Lampante) were investigated to define their possible differences useful for quality discrimination. Total phenolic content discriminated the sample Lampante olive oil (LVOO) with values (0.95 mg GAE/g) significantly lower than the other oils (1.85, 1.80 and 1.98 for A, D and E samples, respectively). The principal component analysis (PCA) revealed that sensory attributes ("bitter" and "pungent") and antioxidant capacity (expressed by FRAP and ABTS$^{\bullet+}$) are positively correlated with Extra-virgin olive oil (EVOO) and Virgin olive oil (VOO) categories, evidencing high values. In conclusion, based on the evaluated parameters, differences between the different olive oil categories were found. Still, they did not allow us to clearly separate the two categories of Extra-virgin olive oil (EVOO) and Virgin olive oil (VOO) oils.

**Keywords:** antioxidant capacity; sensory analysis; quality; olive oil; bioactive compounds

## 1. Introduction

Virgin olive oil (VOO) is defined by the International Olive Oil Council (IOOC) as the oil from the fruits of olive trees, exclusively obtained by mechanical or other physical processes [1]. The IOOC and European Communities Legislation (EC) identified analytical methods and quality parameters such as peroxide value (PV), acidity, Ultraviolet (UV) absorbance values ($K_{232}$ and $K_{270}$), and sensory analysis for VOO to define their commercial grading reflecting their quality [1–3]. Sensory evaluation is one of the most important legal standards to identify olive oil quality and so to differentiate high-quality from low-quality olive oil. Taking this into account, to guarantee the proper classification the IOOC has instituted rules and a specific procedure for the organoleptic assessment and behavior of sensory panels around the world. Through the determination of the defect's intensity, as well as the presence of "fruity", the official panel correctly classifies the oil into its appropriate category [1]. According to the IOOC, based on the results of sensory analysis, olive oils are classified as EVOO, Extra virgin olive oil (the median of the defects is 0 and the median of the "fruity" attribute is above 0); VOO, Virgin olive oil (the median of the defects is above 0 but not more than 3.5 and the median of the "fruity" attribute is above 0); OVOO, Ordinary virgin olive oil (the median of the defects is above 3.5 but not more than 6.0, or the median of the defects is not more than 3.5 and the median of the "fruity" attribute is 0); LVOO, Lampante virgin olive oil (the median of the defects is greater than 6.0). Olive oils classified as Lampante virgin must be refined before being sold [1]. The Extra virgin olive oil is the grade of the highest quality. It must have no defects and greater than zero "fruity" attributes as estimated by a certified taste panel. Between the physicochemical parameters, it must have a free acidity of less than 0.8% and a peroxide value that doesn't exceed 20 milliequivalent $O^2$ kg$^{-1}$ of oil [3]. Several

studies [4–7] demonstrated that EVOO has many beneficial effects on human health such as antioxidant, anti-inflammatory, cardioprotective and anti-tumor properties. Many of these effects are due to its unique chemical composition, as the singular lipid profile, the important source of beneficial antioxidants, and the bioactive compounds [8]. The chemical composition of EVOO depends on the synergy of multiple factors such as the cultivar, the climatic conditions, the harvest time, the production technologies, and the storage conditions [9]. EVOO has been considered for a long time only a dressing, but now, due to its nutritional and sensory features, it is most esteemed within the Mediterranean diet which has been also related to reduced risk of overall mortality [10].

As highlighted above, various clinical data have shown that the consumption of olive oil often gives benefits to human health. High quality EVOO is rich in bioactive molecules demonstrating to have, among other features, a protective capacity against free radicals [6]. In this context, it is important to underline that the high quality of EVOO is linked, in addition to the other compounds, to the antioxidants [11] and its sensory properties. The antioxidant activity in extra virgin olive oil is bonded to the presence of several phenolic compounds: the phenolic alcohols, 3,4-(dihydroxyphenyl)ethanol (3,4-DHPEA or hydroxytyrosol) and the p-(hydroxyphenyl)ethanol (p-HPEA or tyrosol); the secoiridoids, the dialdehydic form of carboxymethyl elenolic acid linked to hydroxytyrosol (3,4-DHPEA-EDA) and the dialdehydic form of decarboxymethyl elenolic acid linked to tyrosol (p-HPEA-EDA), called oleocanthal, the 3,4-(dihydroxyphenyl)ethanol elenolic acid (3,4-DHPEA-EA), which is an isomer of oleuropein aglycon, and the p-(hydroxyphenyl)ethanol elenolic acid (p-HPEA-EA) or ligstroside aglycon. These components are particularly important for sensory positive traits bitter and pungent, in fact, the 3,4-DHPEA-EDA, is responsible for the bitter taste, while p-HPEA-EDA is responsible for the pungent taste [11].

Indeed, considerable differences characterize the different categories of olive oils, regarding both the sensory and nutritional profiles [9]. Therefore, the health claim addition on the label would help to understand its health potential related to its quality. About this, the European Food Safety Authority has introduced the Commission Regulation n. 432/2012 [12,13] concerning the health claims of some compounds in foods having a positive biological effect, such as olive oil polyphenols. Therefore, to date, the following health claim can be added on the olive oil label: "olive oil polyphenols contribute to the protection of blood lipids from oxidative stress", but only for olive oil that contains at least 5 mg of hydroxytyrosol and its derivatives (e.g., oleuropein complex and tyrosol) in 20 g of olive oil. In this regard, it is interesting to underline that the EVOO polyphenol content influences both the health-positive effects and its sensory properties [14]. Therefore, the aim of our study was the evaluation and correlation of antioxidant capacity and sensory analysis of olive oils of four different quality grades: Extra virgin, Virgin, Ordinary and Lampante, to verify if better olive oil quality is related to higher amounts of antioxidants and sensory positive characteristics, ensuring the highest health benefits. The novelty of our research will make it possible to discriminate olive oils of different quality levels and we also hope it will help consumers in their purchasing choices.

## 2. Materials and Methods

### 2.1. Samples and Quality Parameters

Olive oils of four different quality grades: Extra virgin, Virgin, Ordinary, and Lampante have been investigated for their sensory profile, antioxidant activity and some qualitative characteristics (free acidity, peroxides, fatty acid composition, specific extinction K232 and K270 values, and ΔK). Olive oils obtained from a blend of different cultivars (Canino, Leccino and Moraiolo) from the Lazio region (Italy) were analyzed. The olive orchard was located in the municipality of Viterbo (Latium region–Italy. Latitude 42° 25' 23.59'' N; longitude 12° 06' 41.08'' E).

Free acidity (FA, expressed as a percentage of oleic acid), peroxide value (PV, expressed as mEq O2/kg of oil), and specific extinction coefficients at 232 and 270 nm (K232, K270 and ΔK) were analyzed according to EU standard [15–17].

### 2.2. Sensory Analysis

The sensory analysis was performed by the Official Tasting Panel of Viterbo (Italy) according to regulations of the European Union [2] and IOOC [1]. The Panel of Viterbo is officially recognized by the International Olive Oil Council (IOOC) and the Ministry of Agricultural, Food, and Forestry Policies (MIPAAF). The oil samples were analyzed for the intensity of defects and positive attributes ("fruity", "bitter", and "pungent") according to the official method [1]. Oils were randomly submitted to the eight tasters and assembled into tasting sessions of four samples with fifteen-minute breaks between sessions. Each taster wrote down the intensity of different attributes on the profile sheet Figure 1. Lastly, the panel leader entered the data resulting from each panel member into the official computer program. For every descriptor, the median score of the eight tasters of the panel was computed. The oil was then graded using the median value of the defects and the median for the "fruity" attribute [1].

**PROFILE SHEET FOR VIRGIN OLIVE OIL**

**INTENSITY OF PERCEPTION OF DEFECTS**

**Fusty/muddy sediment** ________________________________

**Musty/humid/earthy** ________________________________

**Winey/vinegary acid/sour** ________________________________

**Frostbitten olives (wet wood)** ________________________________

**Rancid** ________________________________

**Other negative attributes:** ________________________________

**Descriptor:** Metallic ☐  Dry hay ☐  Grubby ☐  Rough ☐

Brine ☐  Heated or burnt ☐  Vegetable water ☐

Esparto ☐  Cucumber ☐  Greasy ☐

**INTENSITY OF PERCEPTION OF POSITIVE ATTRIBUTES**

**Fruity** ________________________________

Green ☐     Ripe ☐

**Bitter** ________________________________

**Pungent** ________________________________

**Name of taster:**                    **Taster code:**

**Sample code:**                    **Signature:**

**Date:**

**Comments:**

**Figure 1.** Profile sheet for virgin olive oil sensory evaluation [1].

Sensory evaluation of olive oil for legal classification must be carried out by an official panel composed of a group of between 8 and 12 trained tasters. The selection of tasters, the tasting glasses, the test room and the statistical processing of results must be performed under specific rules [18,19]. The method uses a profile sheet for use by tasters reported in Figure 1.

Each panelist after having smelled and tasted the sample oil shall insert the intensity with which they perceive each of the negative and positive attributes on the 10-cm scale in the profile sheet [1]. The panel leader gathers the profile sheets compiled by each taster and shall enter the assessment data in the computer program indicated by the method for the statistical evaluation of the founded results based on the calculation of their median. The value of the robust coefficient of variation which defines the classification (defect with the strongest intensity and "fruity" attribute) must be no greater than 20.0% [1]. According to the median of the obtained values in relation to the perceived defect with the greatest intensity and the "fruity" median, the oil is graded into different quality categories [1]. It is important to underline that, to guarantee the scientificity of the method, the official panels are annually submitted for an international evaluation by the IOOC in order to examine the repeatability of the results among all the panels [19].

### 2.3. Fatty Acid Profile

The fatty acid profile was analyzed according to the European Union Commission Regulation [20]. Chromatographic analyses were carried out through a gas chromatograph (Thermo-Finnigan, Rodano, MI, Italy), equipped with a FID detector and a SP-2560 fused silica capillary column (100 m × 0.25 mm × 0.20 μm film thickness) with helium as carrier gas at a flow rate of 1 mL/min. The analysis was performed at the following temperature program: 140 °C held for 5 min, then increased at a rate of 4 °C/min to 240 °C, and held for the subsequent 20 min. The total run time was approximately 50 min. The identification of the peaks was performed by comparing the corresponding retention times to those of several standards and quantified as a percentage of the total fatty acids [21].

### 2.4. Antioxidant Capacity

2.4.1. Extracts' Preparation for Polyphenol Compounds and Antioxidant Activity Determination

For the analyses of the TPC and antioxidant activity, samples were extracted according to Olmo-Garcia et al. [22] with some modifications. Briefly, 10 g of EVOO was extracted overnight in the dark with 60 mL of MeOH/water (60:40, $v/v$). Then, the samples were centrifuged at 5000× $g$ (ALC PK121R centrifuge; Bodanchimica s.r.l., Cagliari, Italy) for 10 min at 4 °C. The supernatant was collected and used for the TPC and total antioxidant capacity determination.

2.4.2. Total Phenolic Compounds (TPC) Content

The TPC was determined using the Folin–Ciocalteu standard method as modified by Costantini et al. [23] and adapted for 96-wells plates and an automatic reader (Infinite 2000, Tecan, Salzburg, Austria). Briefly, 30 μL of deionized water, was added to 10 μL of ethanolic extract, 10 μL of Folin–Ciocalteau reagent, and 200 μL of 30% $Na_2CO_3$. After 30 min at RT, the absorbance of the mixture was measured at 725 nm on a Uvikon spectrophotometer (942, Kontron Instruments, Zurich, Switzerland). A gallic acid standard curve was prepared and the results were expressed as mg of gallic acid equivalents (GAE)/g of the sample.

2.4.3. Total Antioxidant Capacity Determination

The total antioxidant capacity was assessed by two antioxidant assays based on different chemical reactions [24]: ferric reducing antioxidant power (FRAP), for assessing the reducing power, and 2,2′ -azino-bis (3-ethyl- benzothiazoline-6-sulfonic acid) ($ABTS^{\bullet+}$) for measuring the free radical scavenger activity, with the methods described as follows.

FRAP assay was performed using the method described by Benzie & Strain [25], which was adapted for 96-well plates and an automatic reader (Infinite 2000, Tecan, Salzburg, Austria). The method is based on the reduction of the $Fe^{3+}$-2,4,6-tripyridyl-s-triazine (TPTZ) complex to its ferrous form at a low pH. Briefly, 160 μL of FRAP assay solution (consisting of 20 mm ferric chloride solution, 10 mm TPTZ solution, and 0.3 m acetate buffer at pH 3.6) was prepared daily, mixed with 10 μL of the sample, standard, or blank,

and dispensed into each well of a 96-well plate. The absorbance was measured at 595 nm at 37 °C after 30 min of incubation. The results were expressed as mmol Fe2+ equivalents/g.

The ABTS$^{\bullet+}$ radical scavenging activity was evaluated by the OxiSelectTM Trolox Equivalent Antioxidant Capacity (TEAC) Assay Kit (ABTS) (Cell Biolabs INC.) following the manufacturer's instructions. The absorbance was recorded at 405 nm in an automatic reader (Infinite 2000, Tecan, Salzburg, Austria). A standard curve for Trolox was prepared and the antioxidant capacity was expressed as μmol of Trolox equivalents (TE)/g.

### 2.5. Statistical Analyses

The analytical evaluations were performed in triplicate. All statistical tests were carried out with the XLSTAT Premium Version 2023 (Addinsoft, Paris, France) software using one-way ANOVA. Tukey's least significant differences test was used to describe statistical differences between means at the $p < 0.05$ significance level. Principal components analysis (PCA) was employed to investigate the relationships between sensory analysis and the antioxidant capacity between olive oils of different quality grades.

## 3. Results and Discussion

### 3.1. Quality Parameters

The quality parameters (FA, PV and specific extinction coefficients at K232, K270 and ΔK) are reported in Table 1.

**Table 1.** Olive oils physicochemical analysis (means ± SD).

| Olive Oil Sample | Physicochemical Analysis * | | | | |
|---|---|---|---|---|---|
| | Free Acidity(% Oleic Acid) | Peroxide Value (mEq O$_2$/kg) | K$_{232}$ | K$_{270}$ | ΔK |
| A | 0.20 ± 0.01 [b] | 12.0 ± 0.6 [c] | 2.308 ± 0.03 [c] | 0.149 ± 0.01 [c] | 0.008 ± 0.001 |
| B | 0.47 ± 0.02 [a] | 16.7 ± 0.6 [a] | 2.714 ± 0.05 [a] | 0.290 ± 0.01 [a] | 0.010 ± 0.004 |
| D | 0.18 ± 0.01 [b] | 10.0 ± 0.8 [d] | 2.080 ± 0.08 [d] | 0.132 ± 0.01 [d] | 0.002 ± 0.000 |
| E | 0.45 ± 0.01 [a] | 15.5 ± 0.5 [b] | 2.545 ± 0.05 [b] | 0.205 ± 0.01 [b] | 0.003 ± 0.002 |

* Physicochemical parameters evaluated according to the EU Commission Regulation [15–17]. Means marked by different lowercase letters are significantly different (Tukey's test, $p < 0.05$). Legend: Olive oils of four different quality grades. A: Virgin; B: Lampante; D: Extra virgin; E: Ordinary.

In this regard, several researchers found that environmental conditions and cultivars are factors that mostly influence the quality parameters [26–28]. The first parameter, FA expressed as % of oleic acid, gives details about the quality of drupes and how they are stored during the time from harvest to milling. In fact, FA indicates the level of triglycerides hydrolysis in the virgin olive oil [29]. In agreement with the EU regulation [15] and IOOC [3], in EVOO the FA content must not reach to 0.8%. In our study, all the samples analyzed satisfied this limit. However, in samples B and E (LVOO and OVOO, respectively), FA values were significantly higher (0.47 and 0.45%, respectively) in comparison to samples A and D (0.20 and 0.18%, respectively). Regards the parameter peroxide value (PV), which gives the oxidation state of the oils, the maximum value acceptable is 20 mEq of O2/Kg for the EVOO category. Our findings reported that all samples showed values below this limit [3]. The lowest value was reported in sample D (10.0 mEq of O2/Kg) which is the extra virgin olive oil and so the sample with the best quality. Our research took also into account the quality parameters since they are required (K232, K270 and ΔK) by the EU Commission Regulations [2,3] for quality grade classification of olive oils. Regarding these important quality parameters, it is important to underline moreover, that the absorbance K232 is linked to conjugated dienes while the absorbency K270 is brought about by conjugated trienes due to oxidative phenomena. Based on the analyzed physicochemical parameters it could be deduced that, among the analyzed olive oils, only samples A and D (FA ≤ 0.8% oleic acid, PV ≤ 20 mEq O2/kg, K232 ≤ 2.50, K270 ≤ 0.22 and ΔK ≤ 0.01) would be graded as EVOO. While samples B and E, having K232 > 2.6 and 0.25 < K270 < 0.30, would be

graded as OVOO [3]. It is important to underline that classification was mainly possible for the sensory analysis of defects since the olive oils' physicochemical parameter, as can be observed, were insufficient [30].

### 3.2. Sensory Analyses

The Official Tasting Panel of Viterbo evaluated the olive oil samples as belonging to the VOO category (sample A), the LVOO category (sample B), the EVOO category (sample D) and the OVOO category (sample E), in accordance with the IOOC regulation [1–3]. The results are summarized in Table 2.

**Table 2.** Olive oils sensory data (the median) and respective quality grade classification.

| Olive Oil Sample | Sensory Analysis * | | | | | | Olive Oil Quality Grade ** |
|---|---|---|---|---|---|---|---|
| | Defect Predominantly | | Other Defects | Fruity | Bitter | Pungent | |
| | Type | Intensity | | | | | |
| A | Fusty/muddy sediment | 3.0± 0.2 [c] | Winey-vinegary | 3.0± 0.6 [b] | 4.0± 0.4 [a] | 4.0± 0.3 [a] | VOO |
| B | Rancid | 6.3± 0.3 [a] | Musty | 0.5± 0.4 [d] | 0.5± 0.5 [c] | 0.5± 0.6 [c] | LVOO |
| D | n.d. | 0.0± 0.0 [d] | n.d. | 4.5± 0.5 [a] | 4.0± 0.7 [a] | 4.0± 0.3 [a] | EVOO |
| E | Fusty/muddy sediment | 4.5± 0.1 [b] | Musty | 1.0± 0.4 [c] | 1.5± 0.4 [b] | 1.5± 0.5 [b] | OVOO |

n.d.: not detected. Means marked by different lowercase letters are significantly different (Tukey's test, $p < 0.05$). Legend: Olive oils of four different quality grades. A: Virgin; B: Lampante; D: Extra virgin; E: Ordinary. * Sensory analysis was performed by trained panelists following the IOC regulations [1–3]. ** Olive oil quality grade classification based on the physicochemical levels and the sensory analysis [1–3,15–19]: EVOO, Extra virgin olive oil: (simultaneously: FA $\leq$ 0.8% oleic acid, PV $\leq$ 20 mEq O2/kg, K232 $\leq$ 2.50, K270 $\leq$ 0.22, $\Delta$K $\leq$ 0.01 and the median of the defects is 0.0 and the median of the fruity attribute is above 0.0); VOO, Virgin olive oil (simultaneously: FA $\leq$ 2.0% oleic acid, PV $\leq$ 20 mEq O2/kg, K232 $\leq$ 2.60, K270 $\leq$ 0.25, $\Delta$K $\leq$ 0.01 and the median of the defects is above 0.0 but not more than 3.5 and the median of the fruity attribute is above 0.0); OVOO, Ordinary virgin olive oil: the median of the defects is above 3.5 but not more than 6.0, or the median of the defects is not more than 3.5 and the median of the fruity attribute is 0.0; LVOO, Lampante virgin olive oil: the median of the defects is above 6.0.

In detail, for samples A and E, the defect perceived with the greatest intensity (3.0 and 4.5, respectively) was "fusty/muddy" sediment, which is ascribable to incorrect management of the olives. It is characteristic of oil obtained from drupes damaged since piled or stored in such conditions as to have undergone an advanced stage of anaerobic fermentation, or of oil that has been left in contact with the sediment in tanks [31]. In Sample B the defect perceived with the greatest intensity (6.3) was "rancid", present in oils that have undergone a process of oxidative deterioration. It is known that oxidation may be enzymatic and/or chemical. Enzymatic oxidation is due to the action of the lipoxidase which binds the oxygen to the unsaturated fatty acids of the triglycerides, this is favored by the cellular lesions of the drupe which allow contact between the oil and the enzyme [32]. Chemical oxidation takes place during the preservation of the olive oil through a free radical mechanism. Anyway, oxidation promotes undesirable chemical reactions that wholly declass olive oil's sensory quality. Sample D was judged as EVOO with the following perception intensities of "fruity" (3.9), "bitter" (4.0) and "pungent" (4.0). It is notable that this agrees with the association of high-quality EVOO with an equilibrated harmony between the three positive attributes. Another particularly noteworthy aspect is that for samples A and D (VOO and EVOO, respectively) the values of the "bitter" and "pungent" positive sensory notes were significantly similar (Table 2). This could be explained by the fact that total polyphenols (TPC) are present in the VOO sample in amounts not significantly different from the EVOO sample as can be seen in Figure 2. Our results also showed the association of EVOO with a fruitier attribute. Indeed, the median fruity intensity in sample D (EVOO) is 4.5 and decreases in intensity as defects increase. The median fruity intensity reduces in samples A (VOO), E (OVOO) and B (LVOO) with values of 3.0, 1.0 and 0.5 respectively. These findings are in line with those of Eid et al. [30] who found that the median of the fruity attribute had a similarly high value in the EVOO sample. The obtained data demonstrated

how important performing the sensory analysis is to validate the correctness of olive oils' quality grade.

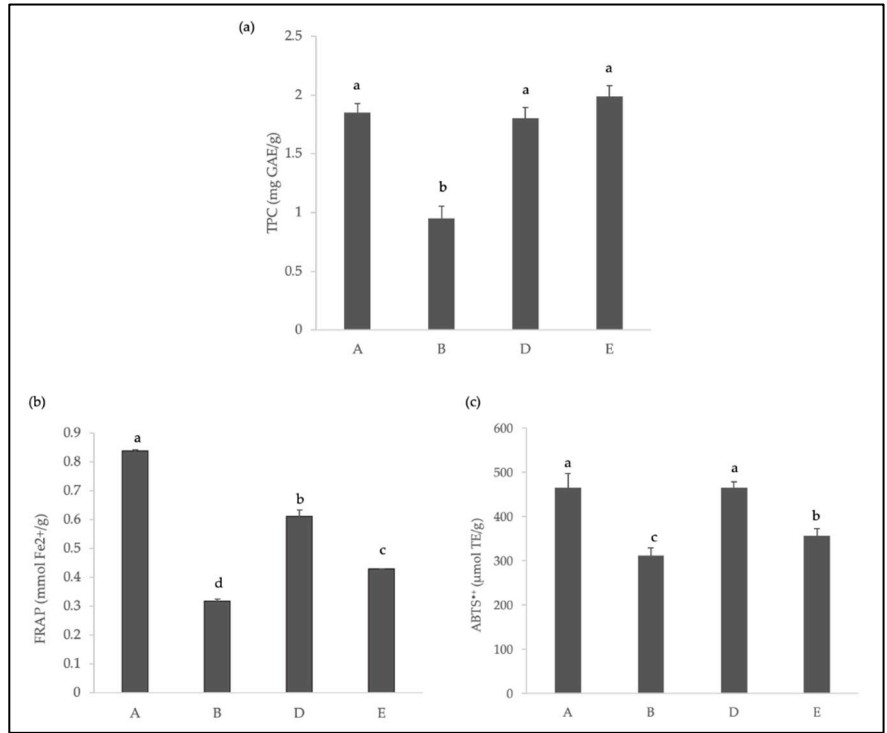

**Figure 2.** TPC (**a**), FRAP (**b**), and ABST$^{\bullet+}$ (**c**) values of olive oils samples. TPC: total phenols concentration. FRAP: ferric reducing antioxidant power. ABTS$^{\bullet+}$: radical scavenging activity assays. Means with different letters are significantly different (Tukey's test, $p < 0.05$). Legend: Olive oils of four different quality grades. A: Virgin; B: Lampante; D: Extra virgin; E: Ordinary.

### 3.3. Fatty Acid Profile

In most of the literature, studies reported fatty acid composition has a great influence on the health benefits of extra virgin olive oil [21,33]. The European Food Safety Authority (EFSA) has also evidenced that unsaturated fatty acids, mainly monounsaturated, help to keep LDL cholesterol at low concentrations in the blood [13]. It is possible to make use of this health claim when unsaturated fatty acids account for at least 70% of total fatty acid content [12]. In samples A, D and E, not only unsaturated fatty acids accounted for more than 80% of the total fatty acid content but, oleic acid on its own featured 75.62, 76.52 and 74.65% of the total fatty acids respectively (Table 3). This evidence showed that in sample D (EVOO) oleic acid represented a higher percentage (76.52%) in comparison to the other samples. This aspect is particularly important since the monounsaturated fatty acid content positively influences the nutritional value and oxidative stability of the oils. In this regard, Reboredo-Rodríguez et al. [34] found that in olive oils oxidative stability is also due to its high levels of monounsaturated oleic acid content. The highest significant percentages were found in sample D (EVOO), followed by A (VOO), and E (OVOO). On the other hand, sample B (LVOO) had the lowest value (69.21%). In Table 3 is possible to observe significant differences between analyzed oils, sample D (EVOO) has the highest value of the oleic acid percentage (76.52%) and the lowest percentage content of linoleic acid (6.87%). On the contrary, sample B (LVOO) presented the highest content of linoleic acid (7.58%) and the lowest content of oleic acid (69.21%). These findings are in agreement with those of Pacetti et al. [33]. As already said, the fact that olive oil composition is characterized by a very high content of oleic acid and low content of linoleic acid determines an olive oil highly strong to oxidation. Several authors [35,36] found that oxidative stability was positively correlated with oleic acid and negatively with linoleic acid. Another important parameter

capable of indicating the quality and stability of olive oils is the C18:1/C18:2 ratio. It is generally accepted that the high C18:1/C18:2 ratio promotes resistance to oxidative decay. Our result shows a significantly highest C18:1/C18:2 ratio (11.13) in sample D (EVOO) assuring a major shelf life by preventing oxidative processes. While sample B (LVOO) presents a lower value (9.13) as reported in Table 3. Anyway, variations in fatty acid composition may also be due to other factors such as cultivars, climatic conditions, and geographical origin [9,29,37].

**Table 3.** Fatty acid composition (%) * of olive oil samples.

|  | C16:0 Palmitic Acid | C16:1 Palmitoleic Acid | C18:0 Stearic Acid | C18:1 Oleic Acid | C18:2 Linoleic Acid | C18:3 Linolenic Acid | C20:0 Arachidic Acid | C20:1 Eicosenoic Acid | C18:1/C18:2 |
|---|---|---|---|---|---|---|---|---|---|
| A | 11.57 ± 0.05 d | 1.16 ± 0.03 a | 3.82 ± 0.02 a | 75.62 ± 0.05 b | 7.22 ± 0.02 c | 0.71 ± 0.01 a | 0.44 ± 0.02 ab | 0.25 ± 0.02 c | 10.47 ± 0.06 b |
| B | 12.03 ± 0.03 a | 1.12 ± 0.01 b | 3.79 ± 0.01 b | 69.21 ± 0.04 d | 7.58 ± 0.04 a | 0.69 ± 0.02 b | 0.45 ± 0.01 a | 0.27 ± 0.02 bc | 9.13 ± 0.06 d |
| D | 11.72 ± 0.01 c | 1.04 ± 0.01 c | 2.55 ± 0.01 d | 76.52 ± 0.04 a | 6.87 ± 0.01 d | 0.67 ± 0.04 c | 0.41 ± 0.01 c | 0.34 ± 0.01 a | 11.13 ± 0.06 a |
| E | 11.97 ± 0.02 b | 1.14 ± 0.02 ab | 2.95 ± 0.04 c | 74.65 ± 0.02 c | 7.41 ± 0.02 b | 0.68 ± 0.01 c | 0.42 ± 0.03 bc | 0.28 ± 0.01 b | 10.07 ± 0.06 c |

* Values are averages of three replicates. Data are mean ± SD. Concentration of fatty acids are expressed in percentage (%), according to official IOC method. Means marked by different lowercase letters are significantly different (Tukey's test, $p < 0.05$). Legend: Olive oils of four different quality grades. A: Virgin; B: Lampante; D: Extra virgin; E: Ordinary.

*3.4. Antioxidant Profile*

3.4.1. Total Phenolic Compounds

Most of the studies in the literature found that phenolic compounds possess antioxidant capacity, nutraceutical effects, and they are also responsible for the positive sensory attributes of bitterness and pungency in EVOO [38–43]. Our results found that the total phenolic compounds in the analyzed samples were in agreement with those of Eid et al. [30]. It is interesting to observe that sample B had a significative difference in TPC with the lowest amount (0.95 mg GAE/g) compared to the other categories (Figure 2). Indeed, according to the sensory parameter limits of IOOC [1] sample B is LVOO category (the median of defects ≥ 6.00), which must be refined before being marketed. The other olive oil categories revealed a higher total phenolic content with values ranging from 1.80 to 1.98 mg GAE/g, in accordance with results by Fanali et al. [44]. In this regard, some authors [40,41] reported that sustained consumption of VOO with high phenolic content was more effective in preserving LDL (low-density lipoprotein cholesterol) oxidation and in raising HDL (high-density lipoprotein cholesterol) levels in comparison to those with low content. Sample E (OVOO) showed the highest TPC amount (1.98 mg GAE/g) supporting what was recently observed by Eid et al. [30]. Indeed, these authors found from in vivo studies in rats that the category of OVOO did not adversely affect the lipid profile of the animals. We would also like to underline that although Olmo-Garcia et al. [22] found higher TPC with the LC-MS method, in comparison to other considered methods (i.e., Folin-Ciocalteau assay, the International Olive Council (IOC) method, and hydrolysis plus HPLC-DAD method) since good correlations were found between the results (R2 > 0.89) we chose the Folin-Ciocalteau, method most commonly used in the literature.

3.4.2. Total Antioxidant Capacity Determination

The antioxidant activities of the different olive oil categories were analyzed using FRAP and ABTS$^{\bullet+}$ assays. Sample B, with the lowest amount of total phenolic compounds (Figure 2), had a lower FRAP (0.32 mmol Fe2+/g) and ABTS$^{\bullet+}$ (311.66 µmol TE/g) values than the other samples (A, D and E) with the highest phenolic amounts. Obtained results showed that the significantly highest antioxidant capacity for the ABTS$^{\bullet+}$ assays, was in samples A and D, 465.91 and 465.70 µmol TE/g, respectively (Figure 2). These samples had also significantly similar TPC values (1.85 and 1.80 mg GAE/g, respectively). Our findings are in agreement with those of Nowak et al. [21] where a positive correlation between the antioxidant activity (determined by ABTS$^{\bullet+}$ assay) and the concentration of total polyphenols was observed. Also, in relation to sensorial characteristics, the analysis

of variance found that no significant differences for "bitter" and "pungent" were found for samples A and D (Table 2). On the other hand, it should be noted that "bitter" and "pungent" sensorial traits have already been reported to be correlated with each other [28]. Moreover, we observed another interesting result: no significant differences in total phenols concentration (Table 2) were found for samples A (VOO), D (EVOO) and E (OVOO). This finding needs further investigation in the future. Indeed, a very recent "in vivo" study [30] investigated, found that, although EVOO had the best effects on health for all the analyzed biological parameters, OVOO did not negatively influence the lipid profile in rats and never manifest any histopathological alterations in examined liver sections. These results could be explained by the high total phenolic content (1.98 mg GAE/g) we found in OVOO (Figure 2), and in accordance with the antioxidant and anti-cancer properties of them [39].

### 3.5. Principal Component Analysis

Principal Component Analysis (PCA) has been conducted to highlight the relationship between antioxidant capacity and sensory analysis of olive oils of different quality grades. The first two components together explained 92.94% of the total variance (F1 83.84% and F2 9.30%) and validated the correlation of antioxidant capacity with the sensory traits of "bitter" and "pungent" (Figure 3). It is evident that the PCA biplot revealed a clustering of the A and D samples in the first quadrant of the plot. Narrow angles reflect variables that are positively correlated with each other, sensory attributes ("bitter" and "pungent") and antioxidant capacity (expressed by FRAP and ABTS$^{\bullet+}$) are grouped close to A and D samples, evidencing high values. Moreover, E and B samples are perfectly separated and located in the third and fourth quadrants respectively. Specifically, B is the sample with the highest intensity of defects compared to E which is located in the inferior part of the third quadrant. However, sample B, belonging to the Lampante category (LVOO) is situated in the fourth quadrant and well distinguished from the others. The F2 component accounted for 92.94% of the total variance in the data set and has been demonstrated to be very useful in interpreting the influence of the presence of sensory defects on the antioxidant capacity and sensory attributes, "bitter" and "pungent", in olive oils of different quality grades. Considering both components, A and D samples are together placed in the first quadrant even if separated from the other samples E and B. This leads to the conclusion that the differences between the different categories of olive oils, based on the evaluated parameters in the present study, do not allow us to clearly separate the two categories of EVOO and VOO oils. These findings are in agreement with those of a recent work [45] in which the authors indicated discrimination between EVOO and VOO and other vegetable oils, but no discrimination between EVOO and VOO was observed.

In Table 4 the eigenvalues, the variability of eigenvalue with the original data (variability row) and the overall variation of the original that all eigenvalues contain (row-Cumulative) are reported. It is possible to observe that the first two eigenvalues comprehend the major part of the information which is included in all indicators (92.938%).

**Table 4.** Calculation of the eigenvalues.

|  | F1 | F2 | F3 | F4 |
|---|---|---|---|---|
| Eigenvalue | 5.019 | 0.558 | 0.357 | 0.051 |
| Variability (%) | 83.642 | 9.296 | 5.944 | 0.854 |
| Cumulative (%) | 83.642 | 92.938 | 98.882 | 99.736 |

Moreover, even though in the PCA emerged a total of six eigenvalues, only four of them are signalized in Table 4 since comprising all the related information (99.736%). This approach made it possible to reduce the number of variables without the miss of significant information.

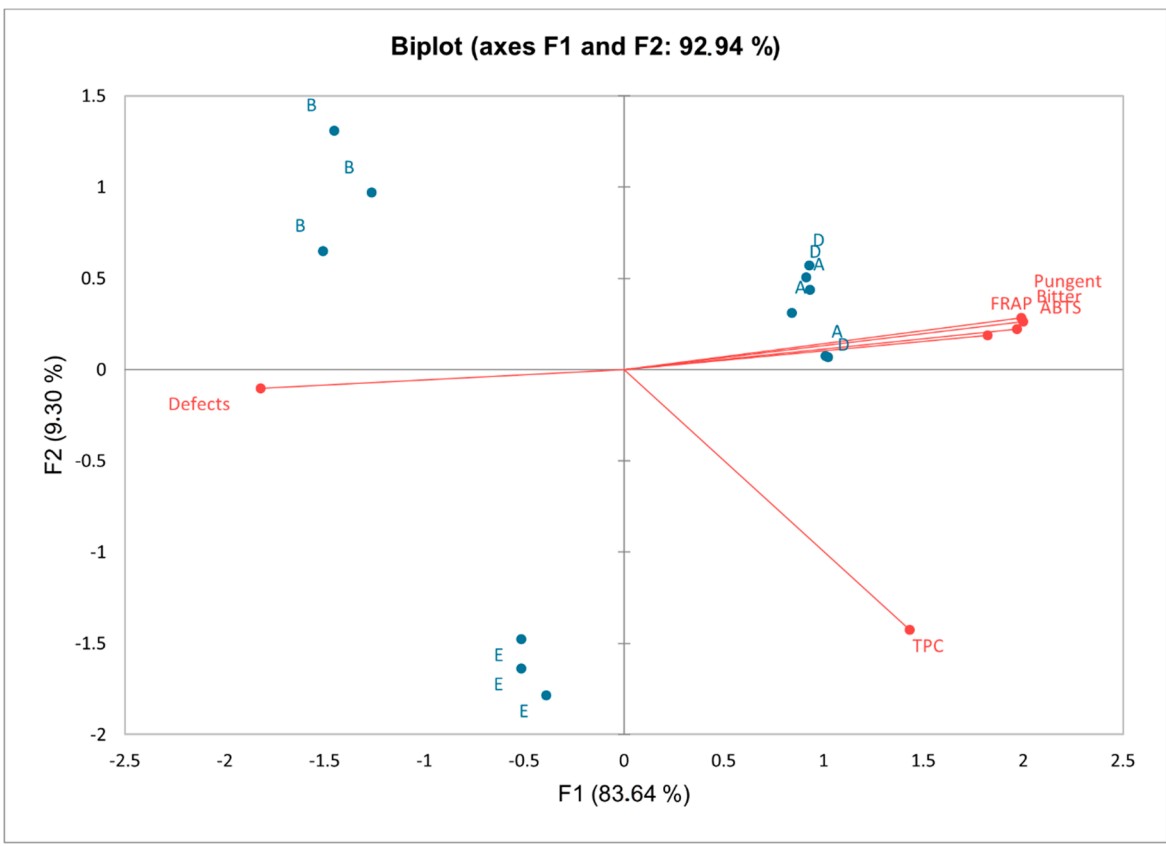

**Figure 3.** PCA biplot performed with the antioxidant capacity and sensory analysis of olive oils of different quality grades. A: Virgin olive oil; B: Lampante virgin olive oil; D: Extra virgin olive oil; E: Ordinary virgin olive oil.

## 4. Conclusions

It is well established that the sensory traits of olive oil are strongly related to its nutritional quality [46,47]. Sensory evaluation of olive oil is conducted by an official panel which, due to its strong regulation, assures that low-quality oils are not being sold as high-quality [1]. Our study represents a preliminary study on the correlation between antioxidant capacity and sensory analysis of olive oils of different quality grades. Based on the evaluated parameters, antioxidant capacity and sensory analysis, we found differences between the different olive oil categories, but not allowing us to clearly separate the two categories of EVOO and VOO oils. These novelty results will set a precedent for future studies about these oil categories. Further investigations are required for the determination of phenolic compositions through chromatographic methods in order to identify the individual phenolic compounds composition linked to the specific antioxidant activities of olive oils of the different quality grades Extra virgin, Virgin, Ordinary, and Lampante.

**Author Contributions:** All authors contributed to the manuscript writing. M.T.F. conceived the presented idea and the writing of the manuscript; L.C. contributed to the implementation of the research; R.M. and N.M. supervised this work. All authors have read and agreed to the published version of the manuscript.

**Funding:** This research received no external funding.

**Institutional Review Board Statement:** Not applicable for studies not involving humans or animals.

**Informed Consent Statement:** Not applicable for studies not involving humans.

**Data Availability Statement:** The data presented in this study are available on request from the co-author, M.T.F.

**Conflicts of Interest:** The authors declare no conflict of interest.

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
