# Peer review of "Antioxidant Profile and Sensory Analysis in Olive Oils of Different Quality Grades"

_agriculture, doi:10.3390/agriculture13050993_

Round 1
Reviewer 1 Report
My comments were sticky on attached pdf file

Author Response
The authors’ response to the reviewers’ comments is highlighted in yellow as well as with yellow evidence all other corrections were done.
We would like to thank the reviewer for providing this valuable comment.

Reviewer 2 Report
The author studied the antioxidant capacity and sensory analysis of different quality grades of olive oil based on principal component analysis method to determine their differences and assist in the quality identification of olive oil. After careful review, the author needs to make modifications according to the following requirements.
1.In the introduction, the introduction of the research background is too brief. It is recommended to add the antioxidant capacity and sensory analysis content of olive oil, and supplement relevant analysis methods as the main component analysis method to lay the foundation. In addition, it is necessary to emphasize the innovation of this study in the last paragraph.
2.In the section 2.5 on data analysis, principal component analysis methods should be supplemented, such as data standardization, dimensionality reduction, and related formulas.
3.The depth of analysis in the sections on sensory analysis in section 3.2 and fatty acid distribution in section 3.3 is insufficient. It is recommended that the author improve this section.
4.In the process of principal component analysis, the proportion of information extraction, eigenvalues, cumulative variance contribution rate, and component load matrix in variables are not provided, and the author needs to carefully add them.
English writing is relatively good, but the grammar of some long sentences needs improvement.
Author Response

(The authors gave the same response as above.)

Reviewer 3 Report
This article describes a study to determine the antioxidant capacity, as well as a number of physico-chemical parameters of olive oils of various varieties and their organoleptic analysis in order to identify their relationship and the ability to predict the positive antioxidant properties of olive oils based on the organoleptic profile. As a result of the study, the following parameters and indicators of the quality of olive oils were determined: organoleptic evaluation, fatty acid profile, antioxidant capacity and content of phenolic compounds.
The work is done well and causes a positive attitude, but there are a number of comments:
1. It is necessary to add the numerical data obtained in the manuscript to the abstract so that the reader can judge its content.
2. In the introduction, a more detailed list of possible substances with antiradical activity found in olive oil should be given.
3. In materials and methods: if samples for analysis were collected in different parts of the Lazio region, then it is necessary to provide the geographical coordinates of those plantings (orchards) where certain varieties were collected to produce the corresponding types of oil.
4. In materials and methods (chapter 2.4): It is not clear how the methods were chosen to evaluate antiradical activity. For example, the authors refer to work number 21 in the bibliography (Antioxidant Properties and Fatty Acid Profile of Cretan Extra Virgin Bioolive Oils: A Pilot Study). However, this study provides a different list of methods for evaluating antiradical activity: DPPH, ABTS, and Folin-Ciocalteu Assay.
The authors of the manuscript did not justify the use of the FRAP method, since it is now known that this method allows you to get results that are not always consistent with the results from other methods (and requires specific conditions for conducting, for example, low non-physiological acidity values, or as shown in the work of Nilsson 2005 - "Any electron-donor substance, even if it does not have antioxidant properties, with a redox potential lower than that of the Fe(III)/Fe(II) redox pair, is capable of distorting the FRAP values, which contributes to incorrect results," or, as shown by other authors, cannot measure the activity of some substances, such as GSH (glutathione)" (doi: 10.1002/mnfr.200400083).
The use of the Frap method is one of the "weak points" of this manuscript, and the authors should be careful in interpreting the values they obtain. This partly explains why exactly the values of antiradical activity varied so much, and the authors do not discuss this in any way (see paragraphs 8 and 9). For limitations of the Frap method, see for example articles: 10.3390/molecules25225267 or 10.1007/s12161-008-9067-7.
Moreover, the original oil samples could contain other substances with antiradical activity, such as chlorophylls, tocopherols, etc., but the authors nowhere provide spectrophotometric analyzes of the original oil samples (for example, in the green-yellow regions of the visible spectrum - to determine the concentration of chlorophyll in original samples).
Tables 1-4: the authors do not explain why there is no sample under the letter C, but there is a letter C above some of the samples. Example in the attached file. Authors either need to add this pattern or remove the C above other values.
5. Lines 188-189: the authors nowhere discuss why they give the absorption coefficients K232 and K270. The authors need to add either reasoning about this, or add data from other people's studies and compare with the same values ​​of other types of olive oil.
6. In Table 1 it is also not clear why the authors put the parameter ΔK - the authors also need to provide explanations about this.

Author Response

(The authors gave the same response as above.)

Round 2
Reviewer 1 Report
My comments were sticky on pdf attached file.

Author Response
We would like to thank the reviewer for providing all valuable comments.
Reviewer 2 Report
I think it can be published.
I think it can be published.
Author Response

(The authors gave the same response as above.)

Reviewer 3 Report
The authors took into account the edits and improved the quality of their article.
Author Response

(The authors gave the same response as above.)
